# Group testing as a strategy for COVID-19 epidemiological monitoring and community surveillance

**Vincent Brault**[1]☯, **Bastien Mallein**[2]☯, **Jean-François Rupprecht**[3]☯*

**1** Université Grenoble Alpes, CNRS, Grenoble INP, LJK, Grenoble, France, **2** Université Sorbonne Paris Nord, LAGA, UMR 7539, Villetaneuse, France, **3** Aix Marseille Univ, CNRS, Centre de Physique Théorique, Turing Center for Living Systems, Marseille, France

☯ These authors contributed equally to this work.
* rupprecht@cpt.univ-mrs.fr

**Data Availability Statement:** All relevant data are within the manuscript and its Supporting information files.

**Funding:** The authors received no specific funding for this work.

## Abstract

We propose an analysis and applications of sample pooling to the epidemiologic monitoring of COVID-19. We first introduce a model of the RT-qPCR process used to test for the presence of virus in a sample and construct a statistical model for the viral load in a typical infected individual inspired by large-scale clinical datasets. We present an application of group testing for the prevention of epidemic outbreak in closed connected communities. We then propose a method for the measure of the prevalence in a population taking into account the increased number of false negatives associated with the group testing method.

## Author summary

Sample pooling consists in combining samples from multiple individuals into a single pool that is then tested using a unique test-kit. A positive test means that at least one individual within the pool is infected. Sample pooling could provide the means for rapid and massive testing for the presence of SARS-CoV2 among asymptomatic individuals. Here, we do not address any diagnostic problems—e.g. how to use a minimal number of tests to obtain an individual diagnostic—but rather focus on population-scale application of pooling. We first quantify the reduction of test sensitivity due to sample dilution and quantify the efficiency of large pools in (i) obtaining precise estimates of the proportion of infected individuals in the general population at reduced costs and (ii) implementing regular large-scale screenings beneficial in the early detection of epidemic outbreaks within communities (e.g. nursing homes or university campuses).

## Introduction

Testing aims at revealing the presence of viral load of SARS-CoV-2 within infected individuals [1, 2]. At date, the most standard mean to reveal such viral load remains the *reverse transcription quantitative polymerase chain reaction* (RT-qPCR) tests [3]. Bottlenecks in the production

**Competing interests:** The authors have declared that no competing interests exist.

of reactants used in RT-qPCR diagnostic testing [4, 5] contributed to the development of alternative techniques that provide a more rapid diagnostic, e.g. lateral-flow antigen and RT-LAMP tests, yet at the expense of a reduced sensitivity compared to RT-qPCR tests.

In the wake of a COVID-19 second wave in Europe and at the current date, several countries have been implementing or are actively considering the implementation of *massive testing*, e.g. Slovakia, whereby repeated nation-wide screenings based on antigen tests occured on week-ends in November 2020 [6]; the Duchy of Luxembourg, whereby a nation-wide screening based on RT-qPCR tests is scheduled for Spring 2021 [7]; the city of Liverpool (United Kingdom) with operation Moonshot consisting in repeated city-wide screenings using a combination of testing strategies.

As COVID-19 infected individuals may be contagious without showing symptoms, tracing is particularly challenging; while individuals showing no symptoms throughout the infection appear to account for only 15% of infections [8–10], pre-symptomatic individuals appear to cause around 50% of infections approximatively [11–14].

Large-scale testing programs aim at addressing such challenge by allowing an earlier identification of asymptomatic and pre-symptomatic carriers [15]. In China, city-wide testing programs were reported in several cities including Wuhan (May 2020) [16] and Qingdao (October 2020) [17]. These cities relied on a technique called *sample pooling*, equivalently called *group testing*. The principle of group testing consists in combining samples from multiple individuals into a single pool that is then tested using a single test—which, in the COVID-19 context, amounts to using a single RT-PCR well and reactive kit. The pool sample is considered to be positive if and only if at least one individual in the group is infected.

Group testing has a long history that dates back to the seminal work by R. Dorfman in 1943 [18] in the context of syphilis detection, see [19] for a review.

Several teams across the world have developed group testing protocols for SARS-CoV-2 infected individuals using RT-qPCR tests. As early as February 2020, pools of 10 have been used over 2740 patients to detect 2 positive patients over the San Francisco Bay in California [20]. Late April, a report from Saarland University, Germany, indicated that positive sample with a relatively mild viral load from asymptomatic patients could still be detected within pools of 30 [21]. Further works suggest that RT-qPCR viral detection can been achieved in pools with a number of samples ranging from 5 to 64 [22–36].

In parallel, the theoretical literature on group testing for SARS-CoV-2 diagnostic is growing at a fast pace [4, 37–42]. Most of the emphasis has been put on the binary (positive or negative) outcome of tests, with little or no regard on the viral load quantification [3]. Moreover, if the possibility of false negatives is sometimes considered, the increase in the rate of false negatives with dilution of samples due to group testing is not often taken into account [43].

In this article, we do not address any diagnostic problems, such as the question of determining optimal strategies to provide individual positive diagnostic using a minimal number of tests as solved by hypercube methods [23, 26, 43–45] and the P-Best algorithm [24]. We rather propose to evaluate pooling strategies in the non-diagnostic contexts of *screening* and *surveillance*, as defined by the Centers of Disease Control (CDC, USA) terminology [46].

In Section II, we propose a group testing protocol that aims at the early detection of an epidemic outbreak in a closed community, such as nursing homes or universities. In the context surveillance, the size of pools is mainly determined by the maximal tolerated sensibility loss induced by the sample pooling process; the optimal pool size predicted according to a diagnostic criteria addressing the minimizing number of diagnostic is ill-defined in such context, and does not provide an adapted answer to the question of determining whether a disease is present or not absent in a community.

In Section III, we provide a mathematical formula to estimate the viral prevalence (i.e. the fraction of positive individuals among the tested population) based on pool results. Indeed, the laboratory in charge of the screening program may not have access to the subsequent diagnosis results from positive pools; within the American CDC surveillance protocol, non-clinical (e.g. veterinary) laboratories are allowed to perform pooling of samples for surveillance screenings but individual diagnosis is to be performed by a clinical laboratory abiding by the Clinical Laboratory Improvement Amendments (CLIA) rules. Some individuals may also refuse to comply to the diagnosis tests. In addition, diagnostic tests performed on positive pools may also turn negative [47]; we see at least 3 possible reasons for such discrepancies: (i) an inherent false-negative risk in diagnostic tests, (ii) a possible time delay between the screening and diagnosis tests (e.g. that could result in positive individuals in the pool turning negative in the diagnostic test) or (iii) the fact that the screening and diagnostic tests may not rely on the same sample collection—with screenings relying on self-collected nasal swabs or saliva collection, while diagnosis tests are most often performed on nasopharyngeal swabs samples with an arguably higher level of sensitivity.

We find that large pools are extremely efficient at estimating the prevalence. Such estimates could serve as a metric to scale prevention measures within a predefined graded response scheme—e.g. in a college university campus, the decision to switch to remote teaching could be triggered once a critical measured prevalence is reached. Surveillance protocols based on sample pooling have indeed been implemented in several universities across the United States, including Duke University [48] and the State University of New York [49]. Similar protocols have been defined for regular surveillance at Liège University (Belgium), as well as at Nottingham and Cambridge universities (United Kingdom) [50]; in the latter, samples are pooled by dormitories; if a pool turns positive, all individuals are requested to undergo isolation as potential case contact; a second diagnostic test is then performed to find the infected individuals [50].

Both Section II and Section III rely on a realistic models for the risk of false negatives induced by sample pooling. Estimating such risk is the objective of Section I, whereby we provide a short description of the RT-qPCR and a statistical model for its study. In Section I.2, we analyse the distribution of viral loads among a series of clinical datasets to estimate the averaged false-negative rates induced by the sample pooling process, assuming a linear dilution and a fixed positivity cycle threshold.

## Results

### I Models for sample pooling in RT-qPCR test

We present a mathematical model of the RT-qPCR test as well as a new censored-Gaussian method to fit distributions of viral load in the population. We apply our results to the estimation of the increased risk of false-negatives due to dilution.

### I.1 Statistical model for the cycle threshold value (fixed a given viral concentration in the sample)

The RT-qPCR technique is a routine laboratory technique used to estimate the concentration of viral material in samples [51]. A RT-qPCR machine typically returns a $C_t$ value, which corresponds to $-\log_2$ of the initial number of DNA copies in the sample, up to an additive constant and measure error. It is measured as an estimated number of cycles needed for the intensity of the fluorescence of the sample to reach a target value (see Fig 1). The main test in the realization of that test are recalled in Box 1.

Combined measures of two viral RNA strands are also recommended [3]. Here we focus on a single RNA strand detection and we do not model here the possible errors at the reverse

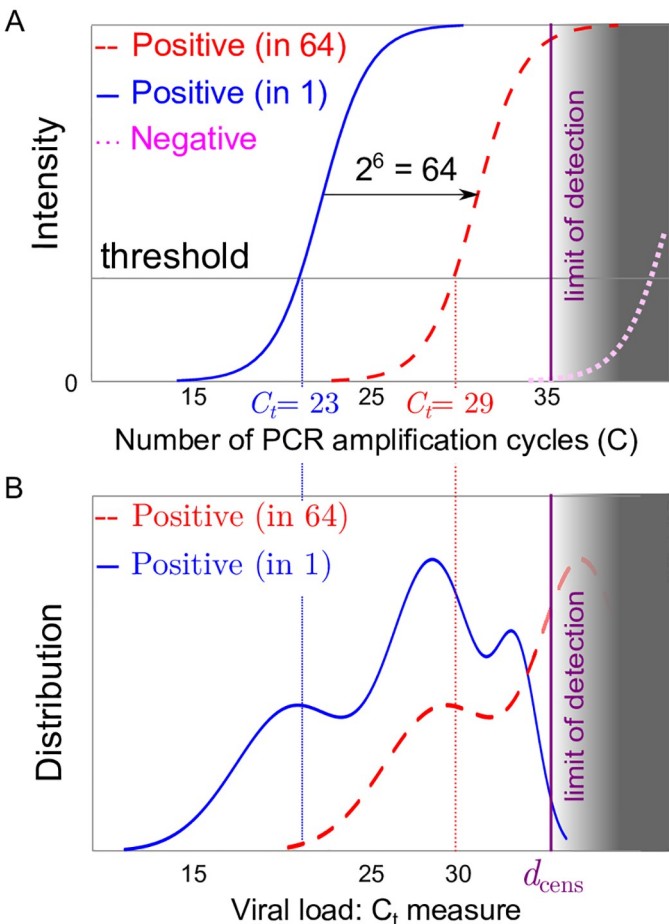

**Fig 1.** (A) Sketch of an RT-qPCR fluorescence intensity signal for a positive patient without pooling (solid red line) a single positive patient in a pool of 64 patients (dashed red curve) and for a negative sample representing the response of an artefact (dotted magenta curve); as pooling dilutes the initial concentration, the pooled response (dashed red curve) is expected to be close to the translation $x \rightarrow x + 6$ from that of a single patient (solid red line). (B) Sketch of the distribution of threshold values for RT-qPCR individual tests (solid blue line) or in pools 64 (dashed red curve); part of the distribution crosses the limit of detection of the test (figured as the grey area) at the detection threshold $d_{cens}$.

## Box 1: A brief description of RT-qPCR tests

We very briefly review some of the steps implemented during an RT-qPCR diagnostic procedure [3]:

1. The sample is treated so that a target RNA sequence (characteristic of the virus) is transcribed into DNA (reverse transcription);

2. The sample is placed in a RT-qPCR machine, which can measure the concentration of DNA of interest in the sample by making it fluorescent;

3. A reactive is added which approximatively doubles the number of DNA of interest at every cycle, driven by temperature changes;

4. The time series of the concentration in DNA over time is recorded; on a linear regression of of the logarithm of the fluorescent signal over time, one deduces an estimate of the viral concentration in the sample from the linear regression value at the origin.

transcription stage, which could lead to some biased measure of the viral load distribution. Depending on the RT-qPCR device, the $C_t$ value of the sample can shift by an additive constant; such constant can be estimated by measuring the $C_t$ value of a standard solution of viral DNA to tare the measure. In any cases, some RT-qPCR device might allow the detection of lower viral loads than others.

**I.1.1 Model of the cycle threshold values for an individual sample.** RT-qPCR tests are prone to amplify non-specific DNA sequences [51, 52] that can trigger an onset of fluorescence in a samples with no viral SARS-CoV-2 load. The fact that such spurious onset of fluorescence typically occurs beyond a relatively large critical number of cycles imposes the following condition on the diagnosis to minimize the risk of *false positives*: a reliable positive result can only be made if the $C_t$ value is lower than a critical value, denoted $d_{cens}$. Here, the onset of fluorescence from virus-free samples will be modelled as if triggered by a vanishingly small artificial concentration, denoted $\epsilon_1$.

We propose to model the number of cycles threshold value $C_t$ as a random variable, denoted by $Y$, that depends on the viral load $c$ in the measured sample as

$$Y = -\log_2(c + \epsilon_1) + \epsilon_2, \tag{1}$$

where we assume that (i) the risk of non-specific amplification (false-positive) $\epsilon_1$ as log-normal distribution with parameters $(v, \tau^2)$; and that (ii) the intrinsic variability in $C_t$ measurement $\epsilon_2$ is a centered Gaussian random variable with variance $\rho^2$.

As mentioned above, tests are considered to be reliably positive when $Y \leq d_{cens}$. To minimize the risk of false positives, the threshold $d_{cens}$ (with cens for censoring) is chosen such that $\mathbb{P}(\epsilon_1 > 2^{-d_{cens}}) \ll 1$. Thus, using that as long as $a$ and $b$ are of different orders of magnitude, we have $\log(a + b) \approx \log(\max(a, b))$, we deduce that

$$Y \approx \min(-\log_2(c), d_{cens}) + \epsilon_2, \tag{2}$$

which obeys the law of a Gaussian random variable with variance $\rho^2$ and mean $-\log_2(c)$, censored at $d_{cens}$.

In the no false-positive risk limit ($\epsilon_1 \to 0$), the RT-qPCR threshold intensity of a negative patient ($c = 0$) would never be reached ($Y \to \infty$ as well as $d_{cens} = \infty$).

**I.1.2 Model of the cycle threshold values for pooled samples.** We now consider what happens when constructing a pooled sample of $N$ samples. For each $i \leq N$, we write $Z_i = 1$ if the sample $i$ contains a viral RNA load with concentration $C_i > 0$, and $Z_i = C_i = 0$ otherwise. In the rest of the paper, we assume that, in a combined sample created from a homogeneous mixing of the individual samples, the viral concentration reads:

$$C^{(N)} = \frac{1}{N}\sum_{j=1}^{N} Z_j C_j. \tag{3}$$

This assumption relies on the fact that infected individuals should have a sufficiently high number of viral copies per sample, so that taking a portion $1/N$ of a virus bearing sample brings a fraction $1/N$ of its viral charge. The result of the RT-qPCR measure of a grouped test with $N$ individuals is then given by Eq 1, with $c = C^{(N)}$, hence reads

$$Y^{(N)} = \min\left[\log_2 N - \log_2\left(\sum_{j=1}^{N} Z_j C_j\right), d_{cens}\right] + \epsilon_2. \tag{4}$$

where $(Z_i, i \leq N)$ are i.i.d. Bernoulli random variables whose parameter is the prevalence of the disease in the population; $(C_i, i \leq N)$ are i.i.d. random variables corresponding to the law of

the viral concentration within samples taken from a typical infected individual in the overall population.

Our model Eq 4 is consistent with the experimental result of [22] as well as [33], whereby linear relations are found between the logarithm of the pool size and the measured $C_t$ that are sufficiently distant from the identified detection threshold.

*Remark* I.1. If it were possible to combine samples without dilution (e.g. following the protocol of [34], whereby the exact same volume of each sample is added to the buffer solution as if the sample were being tested individually), 4 would then be replaced by

$$Y^{(N)} = \min\left[-\log_2\left(\sum_{j=1}^{N} Z_j C_j\right), d_{\text{cens}}\right] + \epsilon_2,\tag{5}$$

in which case, theoretically, pool testing would never loose precision when the pool size increases. However, if the dilution effect occurs for pool sizes exceeding a threshold size $K$, Eq 4 would be replaced by

$$Y^{(N)} = \min\left[\log_2\left(\frac{N}{K}\right)_+ - \log_2\left(\sum_{j=1}^{N} Z_j C_j\right), d_{\text{cens}}\right] + \epsilon_2;\tag{6}$$

where $\log_2(N/K)_+ = 0$ if $N < K$ and $\log_2(N/K))$ otherwise; the analysis would then be similar to what is presented in the rest of the paper, yet with a lower false negative rate.

*Remark* I.2. We expect the RT-qPCR result to correspond to the sample with the highest viral load, up to a dilution-induced drift $\log_2(N)$, under the model hypothesis of Sec I (cf. Fig 1). Indeed, since the viral concentration in randomly selected infected individuals spans several order of magnitudes, we expect that

$$\log_2\left(\frac{1}{N}\sum_{i=1,...,j} C_i\right) \approx \log_2\left(\max_{i=1,...,j} C_i\right) - \log_2(N),\tag{7}$$

for $j$ positive samples with concentration $C_j$ diluted in a pool of $N$. In contrast with [53], we find, based Eq 7, that the measured value of the pooled sample viral concentration cannot be used to estimate the number of infected individual within the pool. However, we point out that the RT-qPCR viral load measure could be used to improve efficiency and cross testing of smart pooling type diagnostic methods, which are beyond the scope of this paper. We plan to investigate this aspect in future work.

In order to determine the statistics of the measured cycle $Y^{(N)}$ in a group test of $N$ individuals, we need a distribution for the value of $C_j$, the viral distribution of infected individuals in the population; this is the objective of the next section.

## I.2 Statistical analysis of the population-level viral load

In this section, we model the $C_t$ distributions extracted from a set of clinical datasets.

**I.2.1 Clinical datasets.** Here we considered four studies in our analysis:

1. The ImpactSaliva dataset [54] providing raw Ct measures from saliva samples (Yale University, USA) with the N1 gene as a target. A set of raw $C_t$ data from $N = 180$ individuals is provided, including 45 $C_t$s values beyond the positivity threshold set at $C_t = 38$.

2. A dataset constructed on a histogram from Lennon et al. [55] based on 2179 nasopharyngeal samples from residents and staff members of nursing homes during the April to May 2020 period; the N1 gene was used as a target (Massachusetts,USA). Up to date and to the

best of our knowledge, Lennon et al. [55] is the largest to study date to present separated $C_t$ histograms between symptomatic ($N = 739$) and asymptomatic ($N = 1440$) individuals at the time of testing.

3. A dataset constructed on a histogram from Jones et al. [56] based on $N = 3598$ nasopharyngeal swabs samples from individuals with various age at La Charité hospital (Berlin, Germany) during the March to early April 2020 period; two target genes (E gene and RdRp) are mentionned in [1].

4. A dataset constructed on a histogram from Cabrera et al. [28] based on $N = 852$ infected nasopharyngeal swabs samples from residents and staff members of nursing homes in Galicia (Spain) during the March to May 2020 period; the Open Reading Frame 1b (ORF1b) was used as a target gene.

As the precise distribution of data points within each bars of the histogram are unknown in the datasets 2,3 and 4, we assume that points were distributed uniformly in their histogram bar class. We have verified the robustness of our fit estimator for several distribution of points which lead to consistent values for our model parameters (see Section I.1. in S1 Text).

**I.2.2 Censored Gaussian model fits.** As we expect the measure error $\epsilon_2$ of the RT-qPCR to be small with respect to the width of the histogram classes, we set $\rho = 0$ in the rest of the section.

**Mixture model**. The shape of the histograms in Fig 2 suggest that the law of the viral load should be distributed according to a mixture of three or more Gaussian distributions. We performed fitting using standard Gaussian distributions models which we refer to as the naive model.

However, as the dataset histograms usually exhibit a sudden drop in the number of detected cases around a $C_t$ value denoted $d_{att}$ (e.g. $d_{att} = 35.6$ for the Jones et al. dataset), that we refer to as the attenuation threshold. We explain these drops by a loss of sensibility of the measure for samples with $C_t$ value between $d_{att}$ and $d_{cens}$ (the limit of detection). We model this loss of sensibility by a fixed probability $q$ of detection above level $d_{att}$.

**Censored models**. To model a partial lack of detection of low viral load ($C_t$ higher than a threshold $d_{att}$), we introduce the partially censored Gaussian variable as a building block for the representation of the density of the viral load in infected patients.

We assume that if the sampled $C_t$ value is lower than the attenuation threshold $d_{att}$; if the value is higher than $d_{att}$, the sample will be detected with probability $q$, and its measure will be registered. Otherwise, it will be discarded as a (false) negative, with probability $1 − q$. The parameter $q$ represents the probability of detection of a viral load that falls below the detection threshold of some RT-qPCR measures.

The assumption that the probability of detection only depends on whether the $C_t$ value is higher than a fixed threshold is of course an important simplification, as one would expect lower viral loads to be more difficult to detect than higher ones. However, the simplicity of this model allows us to study it as a three parameters statistical model, and to construct simple estimators for these parameters. Additionally, it fits rather well the available data, and fitting a more complicate censorship model would require a lot of measures of $C_t$ values close to the detection threshold $d_{att}$. The quantity $d_{att}$ is fixed based on the observed distribution of $C_t$ values in datasets.

To avoid the problem of modelling of the partial censorship, a solution that we implement here as a comparison tool, is *to forget* the values after the threshold and we perform the fit on the *completely censored* model (i.e. with $q = 0$) to the remaining data. See the Method section for further discussion.

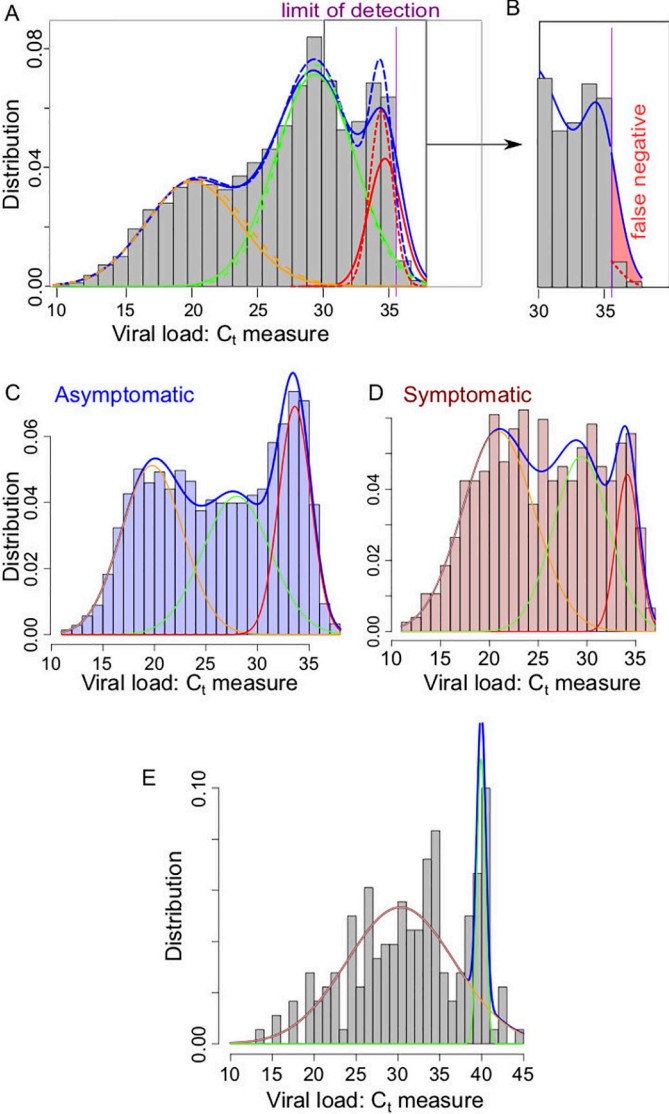

**Fig 2.** (A) Representation of the density for the classical mixing Gaussian model (dashed lines) and the partially censored model (solid lines) each composed as a sum of 3 components for the Gaussian model (orange/green/red dashed lines) and the partially censored model (orange/green/red solid lines); (purple vertical line) location of the threshold $d_{att} \approx 35.6$. Data based on the histogram presented in [56]. (B) Focus on the false negative region, with the estimated false negative probability in the partially censored model (solid line) due to the defect of detection above the threshold $d_{att}$ (red color filled area). (C) Mixing Gaussian model on the ImpactSaliva dataset presented in [54]. (D-E) Mixing Gaussian model on the ImpactSaliva dataset presented in [55] for (D) asymptomatic and (E) symptomatic individuals at the moment of the test. Raw data available in S1 Data.

**Application to the Jones et al. dataset** [56]. We apply here the statistical analysis described in the previous section to simulated data based on the values for the viral load distribution found in [56] with a mixture model and a censoring threshold $d_{att} \approx 35.6$ (so the two rightmost bars in the histogram of Fig 2, that appear much smaller than the nearby values, are supposed to be censored). It is reasonable to assume that the censoring threshold has the same value for each sub-population, as it depends on the test methodology rather than on the tested individuals. In Fig 2, we represent the histogram with the density for the mixture.

We observe that the separation in sub-populations and the resulting densities are very close to the ones obtained in the naive classical Gaussian mixture model, constructed without taking into account the detection threshold. The principal difference between the naive and censored models consists, for the later, in a larger variance that extends above the threshold. To a lesser extent, the sub-population with a median concentration can also exceed the threshold. It is worth mentioning that as expected, the probability of detection below the threshold value is sensibly the same for all three clusters (around 20%).

As a result, using the computed estimates (see Table A in S1 Text) and the model, we can calculate a theoretical false negative rate, see Eq C in S1 Text: in this case, the value is approximately 3.8% (represented by the red area on the Fig 2B); it mostly belongs to the third cluster. Such false-negative estimate remains to be treated with caution.

To validate the censored model, we can verify that if one (i) erases the data to the right of a certain value and (ii) uses the totally censored model on the remaining data, a similar estimate should be obtained for the parameters. We refer to Fig G in S1 Text for the density obtained using the censored mixture estimation with $d_{att} \approx 35.6$, $34.4$ and $33.2$ (removing the first two, the third, then the fourth rightmost bars in the histogram). We observe that the first and second components are globally unchanged. The mean and standard deviation of the last component are almost the same for $d_{att} \approx 34.4$ and $d_{att} \approx 35.6$ (see Table B in S1 Text); only the proportions naturally decrease with the threshold. On the other hand, the mean moves slightly to the left for $d_{att} \approx 33.2$; this is due to the fact that we loose the information of the largest bars of this component. It might also be caused by our ignorance of the exact distribution of $C_t$ values within classes of the histogram (we recall that we assume that it is a uniform distribution).

Note that if we were to set the threshold at $d_{att} \approx 34.4$ as threshold for the partially censored model without erasing data, the optimization procedure `nlm` would not converge. This is further indication that a detection drop happens in the neighbourhood of 35.6.

**Application to the other datasets**. We applied a similar statistical analysis to the other datasets listed in SecI.2.1 and consistently found either two to three sub-populations using our algorithm. The estimation obtained for the Gaussian fit of the $C_t$ distribution they obtained is given in Table A in S1 Text.

In datasets of smaller size [22, 57], the statistical resolution does not allow us to distinguish between several sub-populations; we rather found that the distribution of $C_t$ corresponds to a single Gaussian with standard deviation $\sigma$ in the 5 to 6 range.

**I.2.3 Interpretation of the Gaussian mixture model.** We propose an interpretation of the observed Gaussian decomposition of the log-viral load of individuals based on the viral load temporal evolution within individuals.

**A first model for the individual viral load evolution**. Following [58], we consider a piecewise linear model for the temporal evolution of the mean detected $C_t$ as a function of time after infection $t > 0$,

$$E[C_t(t)] = \begin{cases} \infty & t \leq t_o & \text{(incubation)}, \\ d_{cens} - \frac{\Delta C_{max}}{t_p - t_o}(t - t_o) & t_o < t \leq t_p, & \text{(growth)}, \\ d_{cens} - \Delta C_{max} + \frac{\Delta C_{max}}{t_r - t_p}\left(t - t_p\right) & t_p < t \leq t_r & \text{(decay)}, \\ \infty & t > t_f, & \text{(recovered)} \end{cases} \quad (8)$$

In our first model, we only consider asymptomatic individuals, for which we set $\Delta t_{decay}^{(asymp)} = t_r^{(asymp)} - t_p \approx 7$ days [58]. All other parameters are indicated in Table 1.

**Table 1. Table of values used in Fig 3 for our viral load evolution models.**

| Symbol | Meaning | Date/Value |
|---|---|---|
| $\Delta C_{\min}$ | $C_t$ difference with threshold of the long time plateau | 2 |
| $\Delta C_{\max}$ | $C_t$ difference with threshold of the peak plateau | 13 |
| $t_0$ | Incubation time | Day 2 |
| $t_p, t_{p_1}, t_{p,2}$ | Peak time | Day 5, 5, 7 |
| $t_r$ | Model 1—Decay time | Day 11 |
| $t_r^{(symp)}$ | Model 2—Decay time (symptomatic) | Day 11 |
| $t_r^{(asymp)}$ | Model 2—Decay time (asymptomatic) | Day 14 |
| $t_f^{(symp)}$ | Model 2—End of infection time (symptomatic) | Day 16 |
| $t_f^{(asymp)}$ | Model 2—End of infection time (asymptomatic) | Day 20 |
| $\sigma$ | Noise on the measured $C_t$ | 2 |
| $G(t)$ | Distribution of testing times after infection | N.A. |
| $\tau$ | Decay time in the rate of new infections | 0 or 10 days |

**Distribution of testing times**. For individuals that remain asymptomatic throughout the infection, we expect the testing time distribution to be random, which means the viral load distribution should depend strongly on the rate of new infections.

We denote by $G(t)$ the distribution of testing times $t$ after infection; for asymptomatic individuals, we consider an exponential distribution $G(t) \propto \exp(-t/\tau)$; one may assume that $\tau > 0$ can be approximated by the observed decrease rate of the incidence.

We simulate the viral load measured in a population of $N = 4,000$ infected individuals samples at random times $t_i$, $1 \le i \le N$, distributed according to a model distribution of testing times. We further assumed that the measured viral load law follows a Gaussian distribution $C_i = \mathcal{N}_{d_{\text{cens}}}(E[C_t(t_i)], \sigma)$, $1 \le i \le N$, where $E[C_t(t_i)]$ is given by Eq 8, and $\mathcal{N}_{d_{\text{cens}}}$ is a Gaussian variable conditioned to values inferior to a model limit of detection threshold (set to $d_{\text{cens}} = 35$); $\sigma$ is a constant noise amplitude that models an intrinsic dispersion of the viral load among infected individuals.

Considering the model viral load evolution of Eq 8 (referred to as Model 1 in Fig 3A), exponential $G(t)$ distribution will fail to account the two Gaussian peaks distribution observed in the asymptomatic dataset reported in [55], see Fig 3B.

**Second model for the individual viral load evolution**. To account for the observed peaks in the viral load distribution, we propose the existence of two flat $C_t$ phase that would correspond to the behaviour of the viral (i) near the peak of viral excretion (ii) during a relatively long late infectious phase. Our piece-wise model then reads:

$$E[C_t(t)] = \begin{cases} \infty & t \le t_o & \text{(incubation)}, \\ d_{\text{cens}} - \frac{\Delta C_{\max}}{t_{p_1} - t_o}(t - t_o) & t_o < t \le t_{p_1}, & \text{(growth)}, \\ d_{\text{cens}} - \Delta C_{\max} & t_{p_1} < t \le t_{p_2}, & \text{(peak)}, \\ d_{\text{cens}} - \Delta C_{\max} + \frac{\Delta C_{\max}}{t_r - t_{p_2}}\left(t - t_{p_2}\right) & t_{p_2} < t \le t_r & \text{(decay)}, \\ d_{\text{cens}} - \Delta C_{\min} & t_r < t \le t_f, & \text{(late)}, \\ \infty & t > t_f, & \text{(recovered)} \end{cases} \tag{9}$$

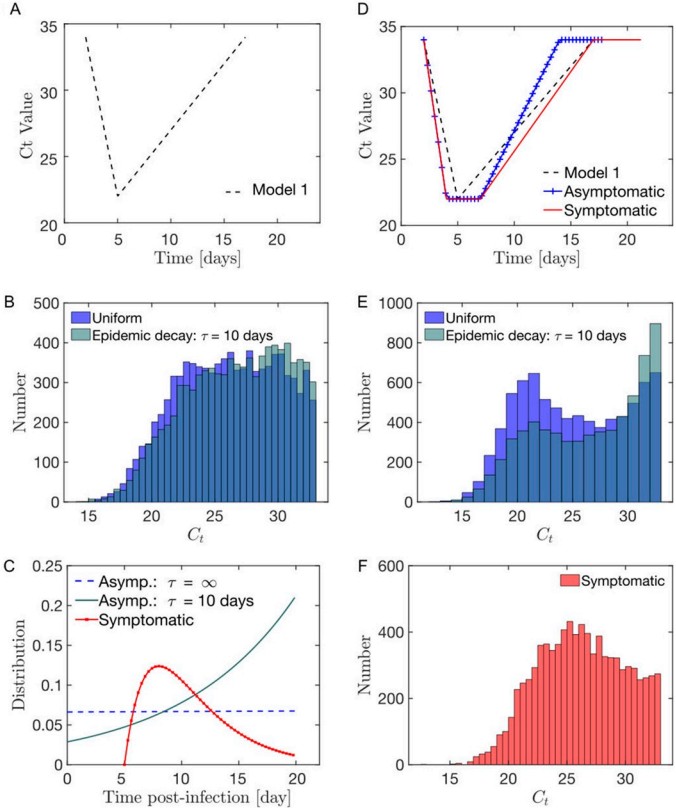

**Fig 3.** (A) Model 1 for the evolution of the viral load post-infection. (B) Modelled distribution in the infection age at the moment of the test for (red) symptomatic individuals as a Gamma function; (blue-cyan) fully asymptomatic (i.e. throughout the infection) individuals either as (blue) a constant if the new infections rate is a constant with time or (cyan) as an exponential if the rate of new infections decays exponentially with time (characteristic decay time $\tau = 10$ days). (C) In the Model 1 context, the distribution of the viral load in asymptomatic individuals is relatively uniform. (D) Model 2 for the evolution of the viral load post-infection distinguishing between symptomatic and asymptomatic (combining [58] and [59]). Parameters estimate are provided in Table H in S1 Text. (E) In the model 2 context, the distribution of the viral load in asymptomatic individuals shows 2 peaks at high and low viral loads. (F) The distribution of the viral load in symptomatic individuals is less bimodal than the observed asymptomatic distribution.

Following [58], we consider different estimates for the decay durations between symptomatic ($\Delta t_{\text{decay}}^{(\text{symp})} = t_r^{(\text{symp})} - t_p \approx 10$ days) and asymptomatic ($\Delta t_{\text{decay}}^{(\text{asymp})} = t_r^{(\text{asymp})} - t_p \approx 7$ days) individuals. We consider the same scaling for the duration of the late infectious phase as the one for the decay time (see parameters in Table 1). The piece-wise model considered in [59] is also nearly flat during the late infectious phase at large time; yet in contrast to Eq 9, [59] considers an instantaneous change of slope at the viral load peak.

**Results**. Based on Eq 9, we find that the viral load distribution for asymptomatic individuals displays two peaks at high and low viral loads, see Fig 3E—in agreement with our analysis of the dataset Lennon et al. [55], see Fig 2D. An exponential decrease in the number of new cases favors the proportion of individual at high $C_t$s, see Fig 3E. The distribution of the viral load in symptomatic individuals is less bimodal than the observed asymptomatic distribution, in agreement with our analysis of the symptomatic dataset from Lennon et al. [55], see Fig 2E.

In [59], a similar results was obtained; a decrease in the incidence rate is shown to be associated to an increase in the proportion of individuals with high $C_t$ value.

Regarding symptomatic individuals, we assume the distribution of testing time to be modelled as a Gamma distribution $G(t) = \Gamma_{\alpha,\beta}(t)$ with parameters $\alpha = 2$ and $\beta = 3$ day$^{-1}$, see Fig 3;

for simplicity, we consider such distribution to be independent of the epidemic status, although a realistic model could include an additional time delay in getting a test during high incidence phases, [60]. Based on both models Eqs 9 and 8, we observe the relatively equally distributed viral load, see Fig 3F; such distribution is in qualitative agreement with the behaviour observed in Fig 2E.

**Multi-Gaussian expression**. Here we interpret the observed multi-Gaussian distribution of the viral load in terms of a simple analytical model. In the absence of noise $\sigma = 0$, the distribution of viral loads corresponding to Eq 9 reads, for any $x \leq d_{\mathrm{cens}}$

$$f_{nn}(x) = \left( \int_{t_{p_2}}^{t_{p_1}} G(t)dt \right) \delta(x - \Delta C_{\max}) + \left( \int_{t_r}^{t_f} G(t)dt \right) \delta(x - \Delta C_{\min}) \tag{10}$$

$$+ \left[ \frac{G(t_1(x))}{t_{p_1} - t_0} + \frac{G(t_2(x))}{t_p - t_{p_2}} \right] \mathbb{1}(x > C_{\min}) \mathbb{1}(x < C_{\max}), \tag{11}$$

where $\delta$ is a Dirac delta-function; $t_1(x)$ and $t_2(x)$ are the two dates such that $C_t(t) = x$, when applicable. In the presence of a noise of amplitude $\sigma(x)$, the measured viral load density reads:

$$f(z) = \mathcal{N} \int_{C_{\min}}^{C_{\max}} dx \int_{-\infty}^{\infty} dy \exp\left( -\frac{y^2}{2\sigma(x)^2} \right) f_{nn}(x - y) \delta(z - x - y), \tag{12}$$

with $\mathcal{N}$ a normalization constant. We therefore expect to obtain a multi-Gaussian distribution for the distribution of viral loads, with two weights being proportional to the time spent at the peak viral phase and at minimal elimination phase for the smallest and largest $C_t$ means, respectively.

**Generality**. Our results are robust to reasonable variations of parameters. We expect our interpretation to be robust to a large class of viral load models that exhibit a sharp viral load rise, plateau at a high level and long decay time.

## I.3 Estimation of the population-averaged false negative rate induced by pooling

**One positive individual within the pool**. The distribution of the viral load of a single positive sample within a pool of several negative samples appears as shifted towards higher $C_t$-values, see Fig 1. A pooled sample returns positive if the average concentration is smaller than $d_{\mathrm{att}}$ with probability 1, or if the average concentration is between $d_{\mathrm{att}}$ and $d_{\mathrm{cens}}$ with probability $q$; thus using the observation of Sec I.2.2, infection will be detected in a group of $N$ individuals typically if at least one individual in the group has a viral load larger than $N2^{-d_{\mathrm{cens}}}$. Therefore, there is a risk that low viral load samples (that would have been tested positive using individual tests) would no longer be positive in pool tests. Similarly to 2, we express the increased rate of false negative due to pooling as

$$\mathbb{P}(-\log_2(C) + \epsilon_2 \geq d_{\mathrm{cens}} - \log_2(N))$$
$$+ (1 - q)\mathbb{P}(d_{\mathrm{att}} - \log_2(N) \leq -\log_2(C) + \epsilon_2 \leq d_{\mathrm{cens}} - \log_2(N)),$$

where $\log_2(C)$ is the viral concentration of the positive individual. For simplicity we neglect the measurement error of the RT-qPCR, i.e. considering that $\rho = 0$, thus an expression for the increased rate of false negatives reads $(1 - \Phi(d_{\mathrm{cens}}^{(N)})) + (1 - q)(\Phi(d_{\mathrm{cens}}^{(N)}) - \Phi(d_{\mathrm{att}}^{(N)}))$, where

$$\Phi(z) = \mathbb{P}[-\log_2(C) \leq z], \tag{13}$$

and $d_{\mathrm{cens}}^{(N)} = d_{\mathrm{cens}} - \log_2(N)$, $d_{\mathrm{att}}^{(N)} = d_{\mathrm{att}} - \log_2(N)$. It is worth noting that a simple upper bound is obtained by setting $q = 0$, i.e. considering that the test is systematically negative when the $C_t$ value is larger than $d_{\mathrm{att}}$ (or in other words, by setting $d_{\mathrm{cens}} = d_{\mathrm{att}}$). This is the choice made when using the completely censored model, and the formula we will use in the rest of the article for the false negative probability is $1 - \Phi(d_{\mathrm{cens}}^{(N)})$.

**Discussion**. The naive and censored fitting models predict lead to two different cumulative distribution expressions $\Phi$. This impact the estimation of the relative false-negative risks defined as $1 - \Phi(d_{\mathrm{cens}}^{(N)})/(1 - \Phi(d_{\mathrm{cens}}^{(N)}))$

- For the Watkins et al. [54] dataset, Fig 4A, our estimate of the relative increase in the false negative is consistent with the quantification performed on pools of a single positive samples in pools of 5, 10 and 20.

- For the Lennon et al. [55] dataset we find that false-negative rate is higher for asymptomatic individuals than for symptomatic ones, see Fig 4, in agreement with a higher proportion of individuals at a low viral load in the former category. In the uncensored model, we make the assumption that the histogram obtained was not subject to any attenuation, while in the censored mode, we consider the sharp drop around $C_t = 35.6$ are being caused by false negative results.

- For the Jones et al. dataset [56], we find that, when estimated by the censored model, the false negative risk function $\Phi(d_{\mathrm{cens}}^{(N)})$ grows quicker as the pool size increases than in the uncensored model, see Fig 4. This is mainly partly due to the fact that the censored model makes the assumption that $d_{\mathrm{cens}} \approx 35.6$, whereas in the uncensored model, the assumption made is that $d_{\mathrm{cens}} \approx 37.3$.

**Multiple positive individuals within the pool**. The case of a pool of $N$ samples that contains $k > 1$ positive individual is particularly relevant as pooling may be achieved on

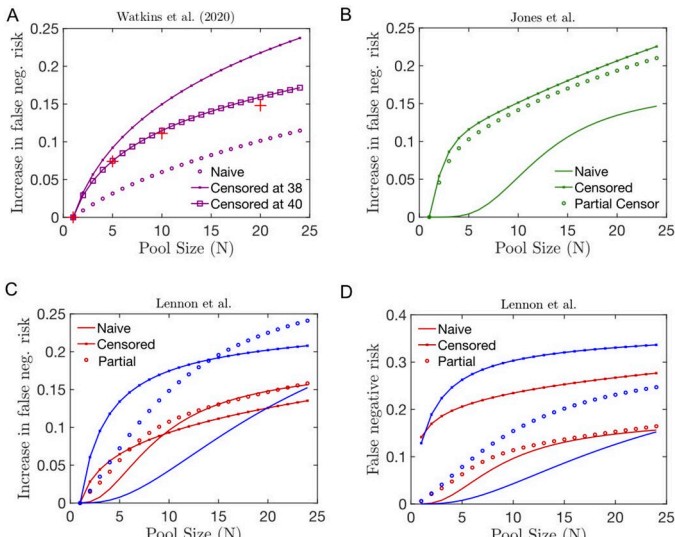

**Fig 4.** (A-C) Relative increase in the false negative risk $(1 - \Phi(d_{\mathrm{cens}}^{(N)}))/(1 - \Phi(d_{\mathrm{cens}}^{(1)}))$ in pools of size $N$ including a single infected individual whose viral distribution is estimated using the naive (solid line), partially censored (circle) or fully censored (crossed line) fitting method of the following datasets from (A) Watkins et al. [54], (B) Jones et al. [56] and (B) Lennon et al. [55]. In (A), we superpose the clinical estimation of the risk of false negative provided in [54] (red crosses). Here, in contrast to [26], we do not change the threshold level of positivity compared to the individual test.

individuals living in the same household, as in [47], or students sharing the same residence hall, as mentionned in [61]; in these cases, the fact that one individual is infected increases the probability that more individuals in the pool are infected as well. Such correlation has been clinically found to lower the risk of false negative risk, an effect coined *hitchhiking* in [47]. In Sec II in S1 Text, we provide estimation for the false-negative rate in pools, we expect to depend on the prevalence among the tested individuals.

Choosing a correct statistical model for the distribution of $C_t$ values has a critical impact on the estimation of the false negative risk, but a less critical one on the estimation of the efficiency of screening strategies, as discussed in the next Result section.

## II Group testing and epidemic outbreak surveillance

We now consider some applications of group testing to the early detection of an epidemic outbreak within a community (with a total number of individuals denoted $A$) that is interconnected and reasonably closed to the outside (e.g. schools, nursing homes, detention centres).

Here, we focus on the false negative risk. False positives are also a concern in low prevalence settings whereby positive predictive value might be low. However, positives appear very rare in RT-qPCR tests—with an estimated higher bound at 0.01% [62].

### II.1 Risk mitigation from a single pre-symptomatic individual

We first consider the impact of group testing strategy, consisting in $k$ group test with pools of $N$ individuals, on the early time of the outbreak $t \ll \lambda^{-1}$. With a unique infected individual in the population, the detection probability reads

$$\mathbb{P}[+|k \text{ tests}] = kN\Phi_0(d_{\text{cens}}^{(N)})/A, \quad \text{with } kN \leq A, \tag{14}$$

where $\Phi_0(d_{\text{cens}}^{(N)})$ is defined according to 13, with the difference that the assumed viral load of the patient 0, corresponding to that measured at early times, may need not be equal to the distribution estimated in 21 based on clinical data. For simplicity, we will assume in the following that $\Phi_0$ is the cumulative distribution of a log-normal viral load distribution $\log N(\mu_0, \sigma_0)$ of mean $\mu_0$ and variance $\sigma_0$.

We first consider of a patient 0 with a weak viral load ($\mu_0 \approx 30$), see Fig 5A. Such low viral load can model the case of a presymptomatic individual, e.g. with a testing time distribution $G(t)$ distributed in the $t = t_0$ to $t = t_p - 2$ days. In Fig 5, we represent the evolution of the probability to detect the patient 0 as a function of the total number of sampled individuals in a population of size $A = 120$. We observe that if $\mu_0$ is close enough to $d_{\text{cens}}$, i.e. if the viral load of the patient 0 is close to being undetectable, then there will exist an optimal size for the pools. When $N$ becomes too large the risk of false negative overcomes the potential benefits of testing larger portions of the community (see Fig 5A). In contrast, if the viral load of patient 0 is slightly higher, the detection probability becomes a monotonic function of the pool size $N$, indicating that larger pools are always beneficial. Additionally, if using multiple tests increases the detection probability when the viral load is close to the detection threshold, using multiple tests has a smaller impact when the viral load gets easier to detect.

Here we first considered the case of a patient 0 with a weak viral load; however, [63] indicates that a large fraction of presymptomatic individuals detected in a nursing homes had relatively high viral loads (with a mean $C_t$ in the $\mu_0 \approx 20 - 25$ range), which tends to indicate that screening methods based on pooling would be even more efficient than suggested in Fig 5A. We then considered the case of single individual with a viral load distributed according to the fits of the datasets Lennon et al. [55] and Jones et al. [56]; in these instances too, we find that

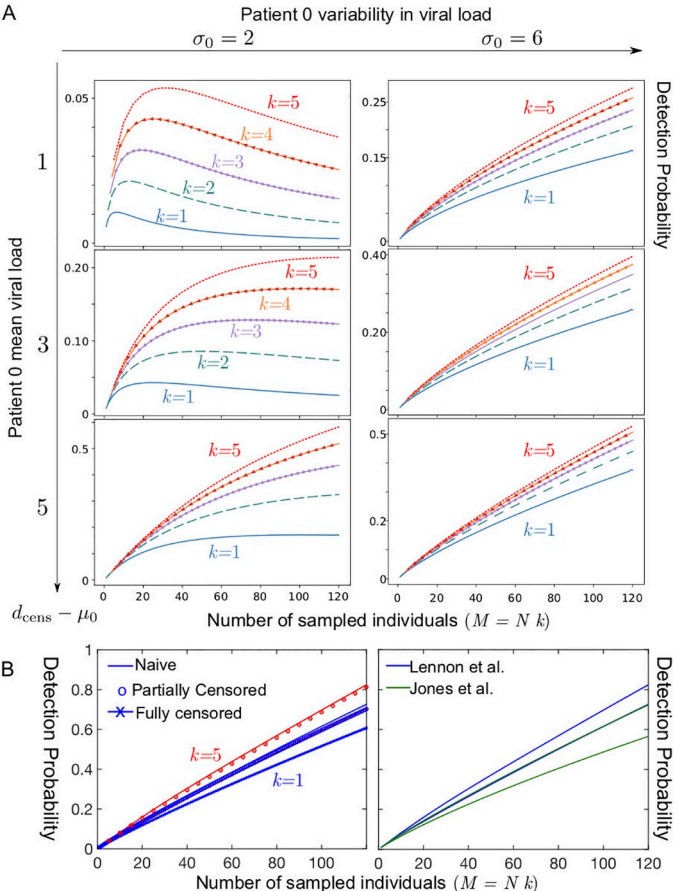

**Fig 5.** Detection probability within a community of 120 as a function of the total number of sampled individuals $M = k \times N$, where $k$ is the total number of tests used and $N$ the number of samples pooled together in a test (A) Case of a single patient 0 with low viral load; $k = 5$ (red dotted line); $k = 4$ (orange line with arrow), $k = 3$ (purple line with circles); $k = 2$ (dashed green line); $k = 1$ (solid blue line) for several values in the parameters describing the viral load of the patient 0 at the onset of contagiosity, expressed in terms of a normal distribution in $C_t$ (the number of RT-qPCR amplification cycles) with a standard deviation $\sigma_0$ and a mean $\mu_0$ and a threshold at a value denoted $d_{cens}$ satisfying: $\mu_0 = d_{cens} - 1$ (top row), modelling a patient 0 with a very low viral concentration, $\mu_0 = d_{cens} - 3$ (middle row), $\mu_0 = d_{cens} - 5$ (bottom row); $\sigma_0 = 2$ (left column); $\sigma_0 = 6$ (right column). (B) Case of a single patient 0 with a viral load distributed datasets(left) for the three fitting methods used to describe the asymptomatic dataset corresponding to Lennon et al. [55], for $k = 1$ (blue) and $k = 5$ (red)and (right) comparing the datasets of Lennon et al. [55] and Jones et al. [56] for the naive fitting method (upper curve $k = 5$, lower curve $k = 1$).

no optimum exists for these two viral load distributions and that large pool sizes are always optimal, see Fig 5B. The parameters used in these computations are recalled in Table 2.

## II.2 Risk mitigation from a cluster of infected individuals

We now consider an epidemic outbreak involving a number $Q$ of infected individuals within a community campus of size $A = 4,000$ at the day $T$ of a screening program. Our objective is to find an estimate of the pool size and testing cost that will ensure detection of at least one individual within the cluster within a maximal tolerated number of days denoted $D$. The probability to detect the outbreak using $k$ pooled tests of size $N$ simply reads:

$$\mathbb{P}(\text{1-day detection}) \approx \left[1 - \binom{A - Q}{Nk} \bigg/ \binom{A}{Nk}\right] \Phi_0(d_{cens}^{(N)}). \tag{15}$$

**Table 2. Table with standard parameter values (with std. the abbreviation of standard deviation).**

| Symbol | Meaning | Value |
|---|---|---|
| $d_{\text{cens}}$ | Maximal cycle number | Table D-E and Table G-I in S1 Text |
| $\mu_i, \sigma_i, p_i$ | Viral load (in $C_t$) distribution fits | Table D-E and Table G-I in S1 Text |
| $\rho$ | RT-qPCR measurement error (std.) | 0 |
| $A$ | Total number in the community | 120 or 4000 |
| $N$ | Pool size | 1–128 |
| $Q$ | Threshold number of infected individuals | 20 |
| $\mu_0; \sigma_0$ | $C_t$-load in patient 0 (mean, std.) | 30–35 |
| $k$ | Number of tests used per day | 1–5 |

In turn, the probability of detection between Day $T$ and Day $T + D$ then reads:

$$\mathbb{P}(\text{D-day detection}) \approx 1 - (1 - \mathbb{P}(\text{1-day detection}))^D. \tag{16}$$

For simplicity, here we considered that $Q$ remains a constant; we do not model the spread of the infection between the day $T$ and the day $T + D$; such spread would only increase the probability of detection, making Eq 16 a lower bound estimate. A more elaborated model exploring the question of the optimal testing frequency the presence of an epidemic spread is discussed in Sec IV in S1 Text.

For the surveillance program to be efficient, detection should be highly probable within a time window should be smaller than the typical time scale of apparition of first symptoms within the forming cluster.

We now consider a reasonable order of magnitude estimation of detection probability Eq 16 with $D = 3$ days and $\Phi_0(d_{\text{cens}}^{(N)})$ estimated using the Jones et al. dataset. With $A = 4,000$ and setting a threshold of $Q = 20$ infected individuals, we find that with $k = 16$ pools of $N = 16$ individuals—i.e. a total number of $N \times k = 256$ sampled individuals per day—the one-day detection probability Eq 15 reaches 72%. The 3-day success probability, as defined through Eq 16 then reaches 99%. With $k = 4$ pools of $N = 16$ individuals, corresponding to 64 sampled individuals per day, the one-day detection probability of Eq 15 is only at 27%; the 3-day success probability, as defined through Eq 16, reaches 62%; yet the 3-day detection probability reaches 85% if the threshold is raised to $Q = 40$ infected individuals.

In the next section, we intend to build an estimator for the prevalence based on the currently available results of pools.

## III Measuring the prevalence using group testing

We investigate in this section the measure of the prevalence of the disease in a population using a group testing strategy. We first consider the assumption of *perfect* tests, i.e. with no risks of false negative nor false positive.

### III.1 Measuring the prevalence in the absence of false-negatives

We assume that we have $n$ pool tests of size $N$ which allow us to sample, at random, $nN$ individuals within a population. Each of these pools is then tested using the perfect tests. For all $i \leq n$, we write $X_i^{(N)} = 1$ if the $i$th test is positive (i.e. if and only if at least one of the $N$ individuals in the $i$th pool is infected), and $X_i^{(N)} = 0$ otherwise. We denote by $p$ the (unknown) proportion of infected individuals in the population. then $(X_i^{(N)}, i \leq n)$ forms an independent and identically distributed (i.i.d.) sequence of Bernoulli random variables with parameter $1 - (1 - p)^N$.

Writing $\overline{X}_n^{(N)} = \frac{1}{n}\sum_{j=1}^{N} X_j^{(N)}$, the quantity $1 - (1 - \overline{X}_n^{(N)})^{1/N}$ is a strongly consistent and asymptotically normal estimator of $p$. Following the seminal derivation proposed in [64] (reproduced in Sec III in S1 Text), one finds that the confidence interval of asymptotic level $1 - \alpha$ reads

$$\mathrm{CI}_{1-\alpha}(p) = \left[ 1 - (1 - \overline{X}_n^{(N)})^{1/N} \pm \frac{q_\alpha (1 - \overline{X}_n^{(N)})^{1/N-1} \sqrt{\overline{X}_n^{(N)}(1 - \overline{X}_n^{(N)})}}{\sqrt{nN}} \right], \tag{17}$$

where $q_\alpha$ is the quantile of order $1 - \alpha/2$ of the standard Gaussian random variable.

The precision of the measure of prevalence decays as $n^{-1/2}$, with a prefactor that depend on the prevalence $p$ and the number $N$ of individual per pool. There exists an optimal choice of $N$ that minimizes the value of this prefactor, largely improving the precision of the measure. Again following [64], one shows that the prefactor in Eq S12 is minimal when the number of mixed samples per pool is equal to:

$$N_{\mathrm{opt}}^{(\mathrm{perf})} = -\frac{c_\star}{\log(1-p)} \Leftrightarrow (1-p)^{N_{\mathrm{opt}}^{(\mathrm{perf})}} \approx 0.20, \tag{18}$$

where $c_\star = 2 + W(-2e^{-2}) \approx 1.59$ and $W$ is the Lambert $W$ function. Specifically, the size of the pools is optimal when approximately 80% of the tests made on the groups turn positive, in sharp contrast with the diagnostic Dorfman criterion [18].

In a recent guideline [65], the European Center of Disease Control presents a seemingly different expression for the prevalence confidence intervals; however, we point out that these estimators for the confidence intervals width become asymptotically equivalent in the limit of a large number of individuals $N$.

If we measure the prevalence of the population using group testing, choosing $N = N_{\mathrm{opt}}^{(\mathrm{perf})}$ for the size of the groups, then measuring with a given precision the prevalence will require significantly less tests than if we were to use one test per sampled individual (i.e. if $N = 1$). On the other hand, using this group testing method increases the total number of individuals needed to be sampled, which also has a cost to be considered. However, one can observe that the bottom of the valley of the (red) functions plotted in Fig 6, that represent the number of tests needed as a function of the size of the pool, is rather wide and flat. There is therefore a large variety of quasi-optimal pool sizes that can be chosen with minimal diminution of the precision in the measure of the prevalence.

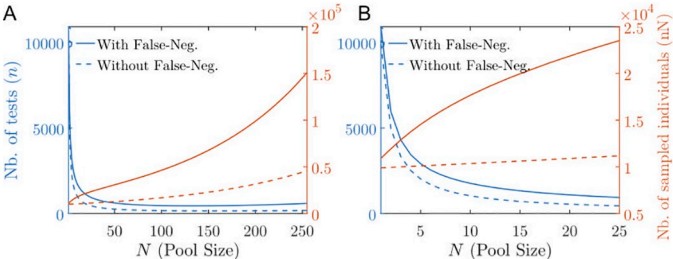

**Fig 6.** (A,B) Total number of tests (red) and total number of sampled individuals (blue) in order to estimate a prevalence of $p = 1\%$ with a ±0.2% precision with 95% confidence interval as a function of the pool size $N$ for the perfect case (dashed lines) with no false negative versus the case with false negatives (solid lines) estimated according to the Lennon et al. asymptomatic dataset [55]. In (A) $N$ ranges from 0 to 25; in (B) $N$ ranges from 0 to 128. The optimal pool size $N_{\mathrm{opt}}^{(\mathrm{perf})}$ is beyond the $N$-axis limit.

**Table 3. Table of the pool size as a function of the number of tests for a prevalence of 3% measured with a precision of 0.2% at a 95% confidence interval, for both perfect tests (with no false negatives, see Sec III.1) and imperfect tests (with false negatives estimated using the Jones et al. dataset; model parameters defined in Table 2); computed using Eqs 17 and 19.**

| Pool size N | Perfect tests | | Imperfect tests | |
|---|---|---|---|---|
| | Number of tests n | Sample size nN | Number of tests n | Sample size nN |
| 1 | 29100 | 29100 | 29464 | 29464 |
| 2 | 14775 | 29550 | 15069 | 30138 |
| 3 | 10003 | 30009 | 10261 | 30783 |
| 5 | 6191 | 30955 | 6411 | 32055 |
| 10 | 3350 | 33500 | 3530 | 35300 |
| 20 | 1973 | 39460 | 2130 | 42600 |
| 30 | 1561 | 46830 | 1716 | 51480 |
| 50 | 1349 | 67450 | 1525 | 76250 |
| 100 | 1884 | 188400 | 2235 | 223500 |
| 200 | 10378 | 2075600 | 13105 | 2621000 |

In Fig 6, we consider the case of a prevalence at $p = 1\%$, in which case the the optimal pool size $N = N_{\mathrm{opt}}^{(\mathrm{perf})}$ is larger than 255. Choosing a pool size of $N = 20$ requires almost a 100% increase in the total number $Nn$ of sampled individuals but more than a 10 fold decrease in the total number of required tests, see Fig 6.

In Table 3, for illustrative purposes, we consider another example of a high prevalence setting ($p = 3\%$) in which expect we expect the optimal pool size (minimizing the number of tests needed) to be in the clinically achievable range $N_{\mathrm{opt}}^{(\mathrm{perf})} \approx 50$ [22]. At optimality, the number of tests needed is divided by 20 as compared to individual testing. In this case, the total number of individuals that need to be sampled is more than doubled compared to individual testing ($N = 1$), see Fig 6. Choosing instead a pool size of $N = 20$ requires almost the same number of tests, yet at a cost of almost a 30% increase in the total number $Nn$ of sampled individuals. The same observation holds for different values of the prevalence, see Fig M in S1 Text.

## III.2 Measuring the prevalence including false negatives

As discussed in 4, we model the concentration of the pooled sample as the average of the individual sample loads; and we assume that viral concentration becomes undetectable below a given threshold. Therefore, creating groups has the effect of increasing the false negative rate, which has to be quantified. We then use this estimation to un-bias the estimator of the prevalence in the overall population based on group testing, and study its impact on the optimal choice of group sizes.

Assuming a false negative rate of $1 - \Phi(d_{\mathrm{cens}}^{(N)})$ in pool testing with groups of size $N$, we observe that $1 - (1 - \overline{X}_n^{(N)})^{1/N}$ (as defined using the notation of Section III) is a consistent estimator of $p\Phi(d_{\mathrm{cens}}^{(N)})$. As a result, the confidence interval constructed for the prevalence $p$ now reads

$$\mathrm{CI}_{1-\alpha}(p) = \left[ \frac{1 - (1 - \overline{X}_n^{(N)})^{1/N}}{\Phi(d_{\mathrm{cens}}^{(N)})} \pm \frac{q_\alpha}{\sqrt{n}} \frac{(1 - \overline{X}_n^{(N)})^{1/N - 1} \sqrt{\overline{X}_n^{(N)}(1 - \overline{X}_n^{(N)})}}{N\Phi(d_{\mathrm{cens}}^{(N)})} \right]. \tag{19}$$

For the numerical applications presented in Fig 6 and Table 3, we consider a viral load $C$ that is distributed according to 21 using the Jones et al. parameter fits. As expected, due to false

negatives, we find that the number of tests needed to reach a given precision on the prevalence is increased; however we this increase is moderate.

In particular, the optimal pool size value, $N_{\text{opt}}^{(\text{imper})}$, that minimizes the number of tests needed to reach a given precision level, is close to the value $N_{\text{opt}}^{(\text{perf})}$, defined in 18.

Similarly, one can observe that using a different distribution with similar mean and variance for $-\log_2 C$ as 21 would lead to moderate changes of the values estimated in Table 3. While modelling of the viral load of an infected individual is crucial to un-bias the estimator of the prevalence via group testing, the practical implementation of such group testing strategy, i.e. the choice of the group size $N$ and the number $n$ of tests to use, is relatively independent of the precise statistical properties of the viral load distribution. We therefore obtained similar results as for the optimal pool size for the prevalence measurement using the viral distribution extracted from the Lennon et al. and ImpactSaliva datasets.

Based on 19, in Box 2 we propose an iterative method to estimate $p$, which, during a survey, allows for on-the-fly adaptations of the pool size.

### III.3 Group testing and Bayesian inference of the prevalence in sub-categories of the population

The viral prevalence may vary significantly among specific categories within the overall population. In particular, a prevalence reaching 5% was measured among the health care workers population in a hospital [66], which we expect to be significantly higher than the estimate prevalence within the general population.

Here we show that we do not specifically need pool samples from individuals from homogeneous categories in order to recover the distribution of prevalence within these categories.

The protocol described in Box 2 can be adapted to study different prevalences in specific sub-populations, provided that the number of individuals of each subpopulation is known for every grouped sample. In Fig N in S1 Text, we evaluate, as function of the number of tests, the credibility intervals on the prevalence within two categories of the population: one at $p_1 = 5\%$ representing 20% of the total population (a value inspired by [66]), the other being at $p_2 =$

---

**Box 2: A protocol of prevalence determination**

We propose the following procedure for the measure of prevalence via group testing:

1. Start from an a priori estimate for the prevalence ($\hat{p}_0$).

2. Based on the value of $\hat{p}_0$, estimate the number $N$ of individuals in the pool that minimizes the total number of tests needed to achieve the estimation of the prevalence $p$ at the targeted precision and confidence interval.

3. Construct a number of $n$ pools containing each $N$ individuals selected at random in the general population, with $n$ the number of tests available for the measure.

4. Count the number of positive tests and compute the average $\overline{X}_n^{(N)}$.

5. An improved estimate of the prevalence then reads: $\hat{p}_1 = 1 - (1 - \overline{X}_n^{(N)})^{1/N}$ (cf Lemma III.1).

Note that this method can easily be adapted into a Bayesian algorithm, with the number $N$ of individuals tested modified at each iteration of the procedure.

0.5% (a value inspired by [67]). More information on this adaptative protocol can be found in the S1 Text.

*Remark* III.1. Note that once a difference in prevalence is noted from the epidemiological study of the general population, testing can be adapted to construct groups containing only members of one subpopulation to attain similar levels of precision for the prevalence of the sub-populations. The prevalence in the general population can then be recovered by averaging the estimators of the sub-populations. The advantage of these adaptative settings is that the existence of a difference of prevalence in populations can be tested before deployment of resources needed to measure them specifically.

## Discussion

We consider the effect of sample dilution in RT-qPCR grouped tests and we propose a model to describe the risk of false negatives as a function of the pool size. We present a procedure to analyse experimental datasets for the viral load of patients. Inspired by the clinical study [56], we expect the statistics of the number of amplification cycles to be well described as a mixture of 2 to 3 Gaussian variables censored at the RT-qPCR sensibility limit. We interpret this decomposition in terms of a simple model for the evolution in the viral load from samples of infected individuals.

We then considered the interest of group testing methods for large-scale screenings in communities. We have used a minimal set of parameters in order for analytical calculations to be tractable. Including more parameters (e.g. considering a time-dependent infection rate or viral load for patients after their infection, graph of relationship within the community) would be needed in order to obtain conclusive results to be used as healthcare guidelines. In this direction, based on stochastic simulations encompassing a large set of parameters, [68] also concludes on the efficiency of group testing in preventing epidemic outbreaks in health care structures.

Several recent papers indicate RT-qPCR tests based on saliva samples are highly-sensitive [69–74]. Saliva collection appears well accepted [75] while decreasing the cost and risks of sample collection. In this context, saliva sample pooling, which demonstrates reduced loss of sensitivity even in large pool sizes [54] and has been massively used in the State University of New York, appears as a promising solution for regular large-scale surveillance programs.

Group testing could provide the means for regular and massive screenings allowing the early detection of asymptomatic and pre-symptomatic individuals—a particularly crucial task to succeed in the containment of the epidemic [14, 58, 76]. We expect group testing for SARS-CoV2 to remain relevant throughout the upcoming vaccination era, in particular as a tool to track the evolution of viral variants.

## Method

Here we clarify the method used to fit the viral load distribution datasets. We define the *partially censored Gaussian model*, denoted by $\mathcal{CN}_{d_{att}}(\mu, \sigma, q)$, with $\mu$ and $\sigma$ the mean and standard deviation of the Gaussian variable before censorship and $q$ the detection probability above the threshold. If we denote by $X$ the random variable, $f_{\mu,\sigma}$ (resp. $F_{\mu,\sigma}$) the density (resp. the cumulative distribution function) of a Gaussian law $\mathcal{N}(\mu, \sigma)$ then the density of $X$ is defined for every $x \in \mathbb{R}$ by:

$$f_X(x) = \frac{f_{\mu,\sigma}(x)}{q + (1-q)F_{\mu,\sigma}(d_{att})} \times \begin{cases} 1 \text{ if } x \leq d_{att}, \\ q \text{ otherwise.} \end{cases} \quad (20)$$

We also define the *fully censored Gaussian model*, written $\mathcal{CN}_{d_{att}}(\mu, \sigma) = \mathcal{CN}_{d_{att}}(\mu, \sigma, 0)$, such that the fitting density is defined for every $x \in \mathbb{R}$ by

$$f_X(x) = \frac{f_{\mu,\sigma}(x)}{F_{\mu,\sigma}(d_{att})} \mathbb{1}_{\{x \leq d_{att}\}}, \tag{21}$$

where $\mathbb{1}_{\{x \leq d_{att}\}}$ is the indicator function equal to 1 if $x \leq d_{att}$, and 0 otherwise. This analysis allows to test several values of $d_{att}$, the fact that estimates of $\mu$ and $\sigma$ remain stable for different values of $d_{att}$ justifies the validity of the censored Gaussian mixture model.

*Remark* III.2. In the absence of censorship (i.e. in the limits $q \rightarrow 1$ or $d_{att} \rightarrow +\infty$), we check that Eq 20 converges to a Gaussian density distribution.

Due to the presence of the cumulative distribution function of a Gaussian law in the denominator in the normalization constant, it is not possible to obtain analytical forms of the parameter estimators. Nevertheless, we can estimate the parameters using an optimization algorithm like the R function `nlm` (available in [77] and in the S1 Code) which implements a Newton-type algorithm. In S1 Text, we provide the proof of a theorem that guarantees the quality of our maximal likelihood estimators.

## Supporting information

**S1 Text. Supporting analysis and proofs document.**
(PDF)

**S1 Code. Codes used in the paper.**
(ZIP)

**S1 Data. Excel sheet with numerical values that were used to generate viral load histograms.**
(XLSX)

## Acknowledgments

We wish to thank the members of the GROUPOOL & MODCOV19 initiatives, particularly Françoise Praz and Florence Debarre who gave us numerous helpful comments. We thank Philippe Hupé for his critical reading; Marie-Claude Potier, Marc Sanson, Agnès Delaunay-Moisan, Jean-Yves Thuret, Catherine Hill and members of the FranceTest collective for insightful discussions on RT-qPCR tests, the interest of saliva samples, and epidemiological surveillance.

## Author Contributions

**Conceptualization:** Vincent Brault, Bastien Mallein, Jean-François Rupprecht.

**Investigation:** Vincent Brault, Bastien Mallein, Jean-François Rupprecht.

**Methodology:** Vincent Brault, Bastien Mallein, Jean-François Rupprecht.

**Writing – original draft:** Vincent Brault, Bastien Mallein, Jean-François Rupprecht.

**Writing – review & editing:** Vincent Brault, Bastien Mallein, Jean-François Rupprecht.

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
