## [Decision Letter · Decision Letter 0]

25 Oct 2020

Dear Dr. Rupprecht,

Thank you very much for submitting your manuscript "Group testing as a strategy for COVID-19 epidemiological monitoring and community surveillance" for consideration at PLOS Computational Biology.

As with all papers reviewed by the journal, your manuscript was reviewed by members of the editorial board and by several independent reviewers. In light of the reviews (below this email), we would like to invite the resubmission of a significantly-revised version that takes into account the reviewers' comments.

We cannot make any decision about publication until we have seen the revised manuscript and your response to the reviewers' comments. Your revised manuscript is also likely to be sent to reviewers for further evaluation.

Sincerely,

Jingyi Jessica Li

Guest Editor

PLOS Computational Biology

Tom Britton

Deputy Editor

PLOS Computational Biology

Reviewer's Responses to Questions

**Comments to the Authors:**

Reviewer #1: Brault and colleagues present a computational framework for pooling samples prior to Sars-Cov-2 testing as a tool to estimate the prevalence of infection within a population. Pooling samples has often been used in the past to detect low-prevalence targets in an efficient manner by conserving tests, and therefore there has been great interest in adapting this to Covid19 testing. The unique insight here is that, in contrast to the usual approach, the authors do not examine pooled testing as a prelude to the testing of individual samples (i.e. for efficient eventual identification of infected individuals) but rather as an end in itself, in order to efficiently provide population surveillance. To do so, they develop a model of test-positivity grounded in apparently reasonable assumptions about test performance and viral load, estimate a simulated distribution of viral loads using a partially-published dataset, and use this to describe how their computational approaches can be adapted as a tool to monitor the appearance and tracking of prevalence within a defined population.

Rigourous assessments of the computational approaches I will delegate to my co-reviewers, and focus instead on the conceptual framework and applicability of findings.

The major weakness of the paper, as the authors implicitly acknowledge, is the use of a simulated dataset of viral densities represented in Figure 3; this was estimated based upon inspection of a pre-print article reporting many thousands of results from Germany, but the exact data were not available for this report, and furthermore that source paper provides little information about how viral positivity and density are correlated with important clinical and demographic variables, such as age, symptomaticity, etc. This has important implications for the applicability of their suggested testing protocols later in the report. Furthermore, what is the authors' hypothesis as to why these values are distributed as they are?

It would be helpful in the Introduction for the Authors to more clearly differentiate their goals from those of others' in the increasingly crowded covid-sample-pooling space. Namely that mentioned above, which is that the goal here is not to provide a tool to optimize pooled "cascade" testing and efficiently find the few infected individuals, but rather for routine population surveillance , presumably to provide a metric by which to deploy population-level controls (i.e. distancing, etc). Describing a roadmap for how these data might be ultimately actionable would sharpen this focus and clarify the distinct aims of this effort.

Very minor commments: "Contaminated" is an odd choice and I suggest infected in its place. Also viral "charge" is non-standard to me, and "load" would be a more common usage.

Reviewer #2: The authors present a modeling study of pooled testing for SARS-CoV-2. The article is not framed well in infectious disease epidemiology and I recommend the authors work with an infectious disease epidemiologist before resubmitting their paper for review. Many of the ideas that the authors present have been discussed at great length in the field of infectious disease epidemiology, particularly around infectious disease surveillance.

1. The framing of the article is odd and fails to account for understanding of infectious disease epidemiology. Although it is true that pooled testing could be used to estimate the prevalence of COVID-19, prevalence of an infectious disease with such high transmission potential is an odd metric to use in surveillance. As soon as the prevalence is estimated, it would be outdated. I recommend reframing the article around a realistic use of pooled testing, either to increase throughput of incident cases or as screening in attempts to find SARS-CoV-2 infections and break chains of transmission. The authors might consider bringing on an infectious disease epidemiologist to help in this regard, as the revisions needed are extensive. The following instances need to be reframed:

a. Lines 2-6. Regularly monitoring the prevalence of the disease does not prevent the onset of an epidemic wave. Prevalence means nothing without incidence, and monitoring does nothing to prevent increasing transmission. Also prevalence is not the metric that would be used to assess the effectiveness of interventions.

b. Lines 20-23. Although it is true that repeated random samples would give a measure of the prevalence of the pathogen, prevalence is not used to monitor infectious disease unless the carriage is long (such as tuberculosis, malaria, or HIV). With SARS-CoV-2 having about a two-week infectious interval, prevalence becomes basically worthless. Mass testing instead, should be used as an intervention to screen and find infections. Pooled testing is also useful to increase throughput for diagnostic testing.

c. In section 1 the authors presume a point prevalence of 3%. A point prevalence of 3% for SARS-CoV-2 would be alarmingly high and would signify that the transmission rate is out of control. I would recommend at the highest using 1%, and if really trying to

2. Another example highlighting the lack of infectious disease epidemiology understanding is in lines 10-13. This metric is called test positivity, and corresponds to the proportion of tests that are positive among treatment-seeking individuals. The term apparent prevalence is not widely used, and should not be continued. Although the principles discussed in this paragraph are correct, the terminology is wrong and confusing. I suggest reframing this paragraph to discuss how test positivity among treatment seeking individuals will overestimate the prevalence of the pathogen in the population.

3. The authors’ focus on the application of group testing to solve the problem of estimating the prevalence is strange. The problems of estimating the prevalence of a pathogen are rooted in who gets tested and how to draw the sample rather than in how the test is conducted. Pooled testing increases throughput, which then can operationalize testing to screen for more infections.

4. There is great value in estimating the false negativity rate among pooled testing. I would have liked to see that particular problem highlighted and improved, with modeling taking into account an estimate of actual prevalence at the time of testing.

Reviewer #3: Review of “Group testing as a strategy for COVID-19 epidemiological

monitoring and community surveillance “

The paper analyses the use of group or pooled testing for detecting

COVID-19 prevalence in a population. It assumes a limited number of

available tests, and looks at the optimal number of individuals to pool

together into each test to achieve tightest confidence interval around

the estimate of prevalence.

This is a solid and interesting paper, and it addresses all the major

problems associated with pooled testing. I have a few minor comments.

1. Line 90, the paper cites a “classical computation”, giving a 2020

reference. If it is a classical computation, I'd cite an earlier source.

The source cited, does not, as far as I can tell, use the same

computation. Instead it uses c_{*}=1. Please give a correct reference,

and even better an actual derivation of the result. This result seems

indeed valid for reducing the confidence interval, though there are

other possible choices for error reduction, such as mean square error,

or, my personal favorite, maximum information. As the paper points out,

the minimum is very flat and as such these are very similar.

2. The paper derives a distribution of C_{t} values, the viral load.

What is not taken into account or mentioned is that this distribution is

not a constant. Each individual goes through a time course of viral

load, and therefore the distribution depends on the time course of the

disease in the population. I don't see a need to use this fact in the

estimation, but it would be nice to mention this fact, and maybe address

how results would differ, if they would. In my opinion all of section 3

detracts from the main message of the paper. I think it should go in a

supplement, just giving the main result of mixture of Gaussian+censure

in the paper for further analysis.

3. Section 4.4.2, optimization of regularity of test, stands somewhat

apart of the rest of the paper. It is a very interesting question, but

the paper only addresses it via simulation. There is also no discussion

of false positives, which are relevant for timing of outbreak.

4. The paper uses “contaminated individual” for “infected individual”.

Usually, I would use contaminated for a false positive. There are also

some other small problems such as “law of the artefact” on line 138 -

I'm not sure what this means, “exemple” on 105. “But to reach similar

level of precision than in single testing“ 107. “the measure is always

made for samples detected as positive “ 240. “We now show how the

previous analysis of the tests used to measure the viral load in

patients can be used to precise the epidemiological monitoring of the

disease in the general population. “ 330. I think these are errors, but

since I'm not a native English speaker, I'm not totally sure. Careful

editing would be good.

---

**Have all data underlying the figures and results presented in the manuscript been provided?**

Reviewer #1: Yes

Reviewer #2: None

Reviewer #3: Yes

PLOS authors have the option to publish the peer review history of their article (what does this mean?). If published, this will include your full peer review and any attached files.

Reviewer #1: No

Reviewer #2: No

Reviewer #3: No
---

## [Decision Letter · Decision Letter 1]

9 Jan 2021

Dear Dr. Rupprecht,

Thank you very much for submitting your manuscript "Group testing as a strategy for COVID-19 epidemiological monitoring and community surveillance" for consideration at PLOS Computational Biology. As with all papers reviewed by the journal, your manuscript was reviewed by members of the editorial board and by several independent reviewers. The reviewers appreciated the attention to an important topic. Based on the reviews, we are likely to accept this manuscript for publication, providing that you modify the manuscript according to the review recommendations.

Sincerely,

Jingyi Jessica Li

Guest Editor

PLOS Computational Biology

Tom Britton

Deputy Editor

PLOS Computational Biology

[LINK]

Reviewer's Responses to Questions

**Comments to the Authors:**

Reviewer #1: As with my initial review, I've limited my focus to the conceptual approach, interpretation, and potential implementation of the work for disease surveillance. I find acceptable and agreeable the author's edits made in response to my suggestions, and furthermore agree with those made in response to the other reviewers as well.

Reviewer #2: The authors have done commendable work revising the article and responding to my edits. I really enjoyed the introduction, and again incorporating false negatives into the modeling is extremely well done. I am satisfied with their response I include a few more revisions that stood out, all minor.

1. Lines 28-30 could be revised for clarity. Completely asymptomatic infections only occur at most in 20% of the population, whereas pre-symptomatic transmission is quite common. We don’t know yet how much asymptomatic carriers actually transmit coronavirus, but we do know that pre-symptomatic transmission is common. I suggest striking this sentence beginning, “particularly challenging…” and having the previous sentence start the new paragraph with China as the example. There are two systematic reviews for proportion of cases that are truly asymptomatic that would be better references – both found about 15%. One’s now a living systematic review at PLOS Medicine: https://www.ncbi.nlm.nih.gov/pmc/articles/PMC7508369/ and the other is at the Journal of the Association of Medical Microbiology and Infectious Disease Canada: https://jammi.utpjournals.press/doi/10.3138/jammi-2020-0030

2. Line 38 should read “seminal” rather than “semina”

3. Line 87 – This local newspaper article is probably the best write-up of how SUNY is doing their massive testing and could be cited to give some credit to Dr. Middleton’s heroic efforts: https://www.syracuse.com/coronavirus/2020/11/how-upstate-medical-university-used-spit-and-grit-to-make-game-changer-coronavirus-test.html

**Have all data underlying the figures and results presented in the manuscript been provided?**

Reviewer #1: Yes

Reviewer #2: Yes

PLOS authors have the option to publish the peer review history of their article (what does this mean?). If published, this will include your full peer review and any attached files.

Reviewer #1: No

Reviewer #2: No
---

## [Editor Report · Decision Letter 2]

20 Jan 2021

Dear Dr. Rupprecht,

We are pleased to inform you that your manuscript 'Group testing as a strategy for COVID-19 epidemiological monitoring and community surveillance' has been provisionally accepted for publication in PLOS Computational Biology.

Best regards,

Jingyi Jessica Li

Guest Editor

PLOS Computational Biology

Tom Britton

Deputy Editor

PLOS Computational Biology

---

## [Editor Report · Acceptance letter]

11 Feb 2021

PCOMPBIOL-D-20-01303R2 

Group testing as a strategy for COVID-19 epidemiological monitoring and community surveillance

Dear Dr Rupprecht,

I am pleased to inform you that your manuscript has been formally accepted for publication in PLOS Computational Biology. Your manuscript is now with our production department and you will be notified of the publication date in due course.

With kind regards,

Alice Ellingham
